# SAGE: A Unified Framework for Generalizable Object State Recognition with State-Action Graph Embedding

Yuan Zang[1]   Zitian Tang[1]   Junho Cho[2]   Jaewook Yoo[2]   Chen Sun[1]

[1]Brown University   [2]Samsung Electronics

https://brown-palm.github.io/SAGE

## Abstract

Recognizing the physical states of objects and their transformations within videos is crucial for structured video understanding and enabling robust real-world applications, such as robotic manipulation. However, pretrained vision-language models often struggle to capture these nuanced dynamics and their temporal context, and specialized object state recognition frameworks may not generalize to unseen actions or objects. We introduce SAGE (State-Action Graph Embeddings), a novel framework that offers a unified model of physical state transitions by decomposing states into fine-grained, language-described visual concepts that are sharable across different objects and actions. SAGE initially leverages Large Language Models to construct a State-Action Graph, which is then multimodally refined using Vision-Language Models. Extensive experiments show that our method significantly outperforms baselines, generalizes effectively to unseen objects and actions in open-world settings. SAGE improves the prior state-of-the-art by as much as **14.6%** on novel state recognition with less than **5%** of its inference time.

## 1   Introduction

If we turn the recipe book *100 Ways of Cooking Eggs* (Filippini, 1892) into videos, can a modern computer vision algorithm successfully understand all of them? The answer to this question is not as straightforward as it might appear. On one hand, the same objects can exhibit a vast range of visual appearances and physical states (e.g., *a whole egg* versus *a scrambled egg*), especially when human actors interact with them. On the other hand, off-the-shelf detection and segmentation systems tend to focus on high-level object categories, largely overlooking their underlying physical states and dynamic transformations. We aim to develop a unified framework to jointly recognize object physical states and their temporal evolutions from visual cues, by learning from unlabeled instructional videos. We believe that understanding object states offers a powerful, object-centric abstraction for modeling world dynamics. This understanding is essential for structured video comprehension of objects, actions, and skills, and for enabling robots to perceive, predict, and plan when interacting with the physical world, thus providing a promising pathway for robots to learn from human behaviors.

A natural initial approach to achieve fine-grained understanding of object states might be to leverage vision-language models (VLMs), which provide detailed language descriptions for visual data from which object states could potentially be extracted. However, Newman et al. (2024) recently demonstrated that a naive application of state-of-the-art VLMs often fails to adequately recognize object physical states. Alternatively, prior work has explored the use of instructional videos, which are rich in object state transformations, to enhance model training. However, these approaches often model object states as discrete categories (Souček et al., 2024), or train specialist models conditioned on actions (Souček et al., 2022; Xue et al., 2024), making it challenging for them to generalize to novel states belonging to unseen objects or actions. This highlights a critical need for models that can learn a more flexible and generalizable representation of object states and their dynamics.

39th Conference on Neural Information Processing Systems (NeurIPS 2025).

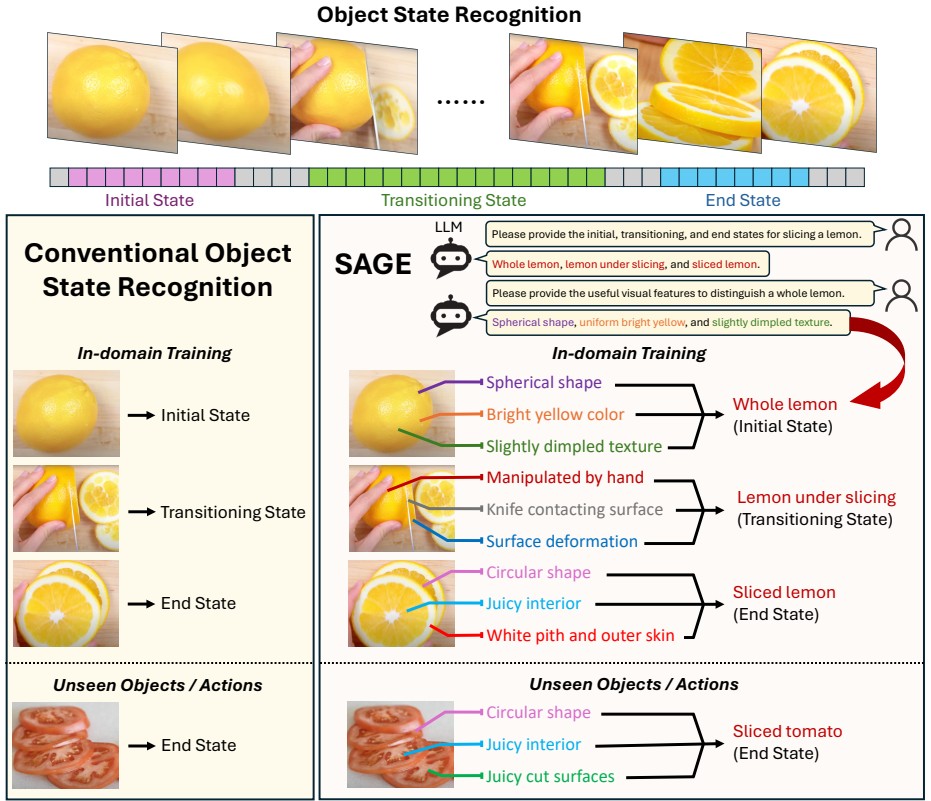

Figure 1: Illustration of state-action graph embedding (SAGE) construction and its application for object state recognition: Given an action, we ask LLM to describe the initial, transitioning, end states and their associated visual concepts. Whereas prior work recognizes states for specific actions, we recognize object states and actions through visual concepts in SAGE. It enables our model to generalize to unseen objects / actions which share similar visual concepts with known ones.

We introduce SAGE (**S**tate-**A**ction **G**raph **E**mbeddings), a framework that leverages multimodal pre-trained knowledge, as in VLMs, yet is able to learn effectively from unlabeled videos, a strategy adopted by aforementioned work on object state recognition. Our key inspiration is that in order to achieve generalizable object state recognition, the model needs to be specific about how a physical concept is rendered visually. As illustrated in Figure 1, SAGE decomposes an object state into a collection of fine-grained visual concepts with language descriptions. Some of them (e.g., *juicy interior*) are shared across objects and actions, facilitating *generalization*, others (e.g., *white pith*) are unique, capturing fine-grained *nuances* of various physical states for the same objects. In SAGE, individually, each action node (a verb-object pair) is connected to three types of state nodes: initial, transitioning, and end states. Each of these state nodes is, in turn, connected to a set of visual concept nodes that describe it. When considered together, visual concept nodes shared by different actions become connected, forming a comprehensive graph of visual concepts, states, and actions. We embed each concept node with multimodal knowledge from VLMs, where visually similar concepts are embedded nearby, even when some of them are unseen during training. An action node is represented by the direction from the initial state to end state in embedding space. We construct the initial SAGE graph using a pre-trained Large Language Model (LLM), allowing nodes for novel actions or objects to be added automatically. We then refine this graph based on multimodal information from a VLM (*e.g.*, assessing if a concept is visually recognizable) and by prioritizing concepts that are shared across a greater number of actions. Once constructed, we train a video transformer model to predict the text embeddings of these visual concepts from video frames, which are subsequently decoded into a sequence of object state and optionally action predictions.

We conduct comprehensive experiments to evaluate our approach on existing object state recognition benchmarks. To make our observations scientifically rigorous, we carefully re-implement the baseline approaches using the same pre-trained vision encoder, and also compare with their reported results

for reference. We evaluate our method on ChangeIt and HowToChange benchmarks with known and novel objects. To further evaluate the model generalizability, we introduce a more challenging setup where both the action and the object in the video are unseen during training. We perform extensive ablation studies showing the contributions of individual design choices in SAGE. Our method demonstrates strong object state recognition performance, especially when generalized to novel objects and actions. Notably, SAGE outperforms the prior state-of-the-art (Xue et al., 2024) across all benchmarks, yielding as much as 14.6% relative precision improvement on state recognition for objects and actions unseen during training, and requiring less than 5% of its inference time when action is not provided during evaluation. Our project website is `https://brown-palm.github.io/SAGE`.

## 2 Related Work

Object states are defined as the physical and functional properties of objects (Liu et al., 2017; Newman et al., 2024). Understanding them allows models to capture the compositionality of objects and their attributes (Misra et al., 2017; Isola et al., 2015; Purushwalkam et al., 2019). Recognizing and localizing object states is essential for video understanding, as objects tend to exhibit an even broader range of state variations (Filippini, 1892), and actions can often be represented as state transformations (Wang et al., 2016; Fathi and Rehg, 2013; Alayrac et al., 2017). Additionally, object states provide valuable cues for skill determination (Doughty et al., 2018) and goal completion (Deng et al., 2020), which are crucial for real-world tasks such as robotic manipulation (Gao et al., 2024).

Prior work has introduced foundation models (Radford et al., 2021; Alayrac et al., 2022; Jia et al., 2021; Wang et al., 2022; Xu et al., 2021) for vision-language understanding, pre-trained on large-scale image/video and caption datasets to learn a unified representation of visual features and textual information. These models have demonstrated remarkable performance across various visual recognition and reasoning tasks. However, they struggle with recognizing object states in images (Newman et al., 2024) and videos (Souček et al., 2022; Xue et al., 2024), as their training objectives often neglect object state transformations. To recognize object states in videos, prior work (Souček et al., 2022, 2024; Xue et al., 2024) has proposed to train classifiers to predict the object states for each frame. However, these methods usually require knowing the action or object information during training and struggle with generalizing to novel actions or objects due to lacking unified representation of object states. We solve these issues by proposing State-Action Graph Embeddings to jointly represent object states and actions via visual concepts.

Visual concepts which represent the primitive features (*e.g.*, colors) of objects have been widely utilized in visual computing. The visual concepts can enable visual models generalize compositionally (Farhadi et al., 2009; Nagarajan and Grauman, 2018; Stein et al., 2024) and enhance their interpretability (Koh et al., 2020; Espinosa Zarlenga et al., 2022). Previous research (Menon and Vondrick, 2022; Pratt et al., 2023; Zang et al., 2025) demonstrates that pre-trained VLMs, such as CLIP (Radford et al., 2021), can learn visual concepts and perform zero-shot recognition based on them. The concepts can be discovered by pre-trained LLMs (Yang et al., 2023; Zang et al., 2025). In this work, we explore how visual concepts can be leveraged to represent object states.

## 3 Method

As illustrated in Figure 2, the input to our framework is a sequence of uniformly sampled video frames encoded by a pre-trained, frozen vision encoder. The action (a verb-object pair) that causes the object state transition can be provided as input (as used by prior work), or otherwise predicted by our framework. The outputs are categorical predictions for all frames, each of which belongs to one of initial state, transitioning state, end state, and background. Unless otherwise specified, we follow the standard setup and assume that each video contains a single action, which can be relaxed when temporal action localization is applied as a pre-processing step.

### 3.1 Base Model

We first contextualize the encoded visual embeddings $v_t$ from all video frames by first linearly projecting them to obtain $h_t$, which are fed to a temporal Transformer to produce the output embedding $Z_t$. An optional token $h_{\text{CLS}}$ is reserved for action prediction and transformed into $Z_{\text{CLS}}$. All outputs are projected into the discrete state / action spaces, where cross-entropy losses are computed on

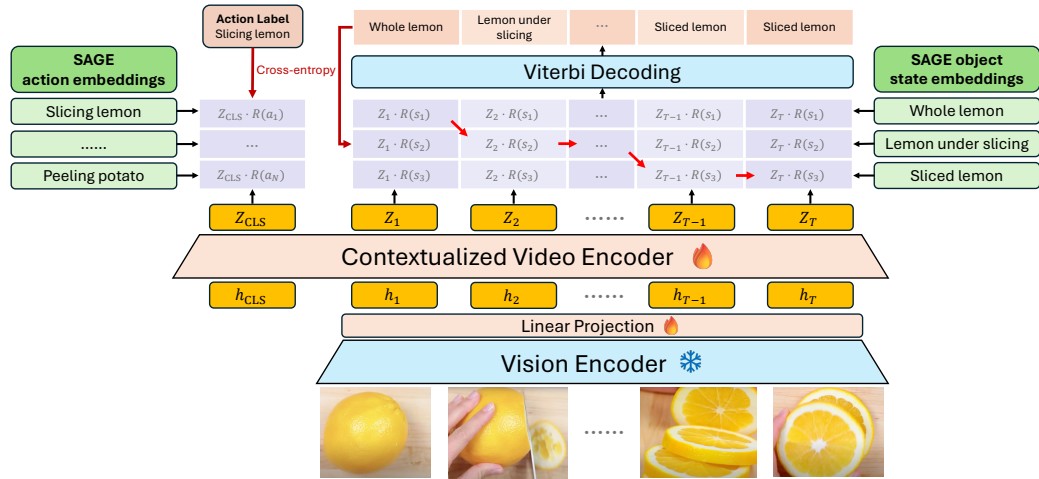

Figure 2: Overview of the training pipeline: Video frames are first encoded individually with a pre-trained and frozen vision encoder, which are then contextualized with a temporal Transformer. We decode the object states by measuring the cosine similarities of the predicted embedding $Z_t$ and the possible state or action embeddings from SAGE.

all frames and the action prediction for model training. Frames that deemed as not containing the object of interest by a VLM are filtered out as background. The overall training objective of our model is a weighted sum of the object state recognition loss and action recognition loss across all non-background frames, *i.e.*, $\mathcal{L} = \mathcal{L}_{\text{state}} + \alpha \mathcal{L}_{\text{act}}$.

**Specialist versus Generalist:** As illustrated in the left side of Figure 1, a specialist model is trained to predict the object states for specific actions or objects, hence not generalizable to novel actions by design. A user needs to specify the input action to select which specialist model to use. A generalist, on the other hand, is trained to predict all object states, even for those unseen during training. This may be implemented by treating initial/transitioning/end states for all objects equivalently, which ignores the significant inter-class variations; also alternatively, by dynamically constructing the output projections (i.e., by concatenating template embeddings of all possible object states) given the action information, which is adopted by our proposed framework.

**Learning from Unlabeled Videos:** While each video in the training dataset is paired with a video-level action label, there is a lack of frame-level annotation necessary for calculating the losses. Following the standard convention, we estimate the noisy "pseudo" object state labels by computing the cosine similarity between frame visual embeddings and state text embeddings using a pre-trained, frozen VLM. We further propose to refine the pseudo labels by applying a temporal constraint (i.e., initial state → transitioning state → final state), which is implemented as a constrained Viterbi decoding algorithm (Viterbi, 1967). The decoding process is implemented through dynamic programming with restricted state transitions.

### 3.2 State-Action Graph Embeddings

We introduce State-Action Graph Embeddings (SAGE) to dynamically construct the state and action embeddings used by the base model to decode object states and actions for generalizable recognition. SAGE first decomposes an action into a state transformations, it then describes each state with a collection of visual concepts, forming a tree structure. Intuitively, we want to advocate the selection of visually distinctive concepts, which capture the nuance of fine-grained object states visually, as well as concepts shared by multiple object states, which facilitate the generalization towards unseen objects. The concept nodes shared by multiple states are merged, forming a densely-connected graph. In addition, both the state nodes and action nodes should be embedded, so that they can be used for the base model for prediction.

**Graph Construction:** To construct a vocabulary of object states, we query an LLM to identify the initial, transitioning, and end states associated with each action in the dataset. For example, the three object states involved in "slicing lemon" include "whole lemon", "slicing lemon", and "sliced lemon".

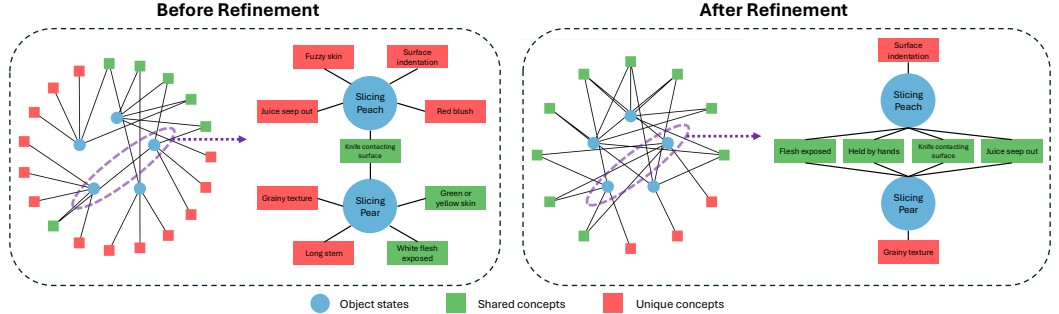

Figure 3: The graph structure of SAGE before and after refinement. We zoom in on a subregion of the graph for demonstration. After refinement, the graph includes more shared visual concepts.

To extract the visual concepts of the object states, we leverage the LLM again to generate descriptive attributes for each object state. Given an object state $s$ and a prompt such as "What are the useful visual features to distinguish a {state name} {object name}?", along with in-context examples, the LLM generates a set of visual concepts $\mathcal{C}_s$ that characterize the object state.

**Node Embedding:** Instead of directly encoding the object state names and action names, we first derive the embeddings of all the visual concepts and then leverage them to compute embeddings for the object states and actions. Ideally, visually similar concepts should be embedded nearby in the text space, further enhancing the generalizability of our framework. As such, we first encode each visual concept as a distributed embedding using the text encoder $E_{text}$ of a pre-trained VLM. We then represent an object state as the average of its visual concept embeddings, *i.e.*, $R(s) = \frac{1}{|\mathcal{C}_s|} \sum_{c \in \mathcal{C}_s} E_{text}(c)$. Since visual concepts are shared across different object states, this formulation provides a unified embedding space, enabling our model to generalize across various object state transformations, including unseen states. Inspired by prior work on relation encoding (Bordes et al., 2013) and state-based action recognition (Wang et al., 2016), we compute the action embeddings as the difference between the embeddings of its end and initial object states, which naturally accounts for object-specific variations, as the same action might have different visual features for different objects (e.g. "boiling egg" and "boiling pasta").

By integrating object states and actions within the state-action graph, our approach captures both their visual attributes and dynamic transformations. The proposed SAGE embeddings enable a structured representation of actions while preserving the compositional nature of object states. This facilitates a more generalizable understanding of object state transitions across different actions.

**Multimodal Graph Refinement:** We propose refining the graph structure to incorporate visual concepts that are reliably recognized by the model to enhance object state recognition accuracy, as well as concepts that are commonly shared across different object states to improve generalizability. Given an object state, we first generate an over-complete concept list with an LLM. We then rank these concepts based on their VL similarity scores with the video frames containing the object state. For a concept $c$, the score is calculated as $\frac{1}{|F|} \sum_{f \in \mathcal{F}} \cos\_\text{sim}(E_\text{text}(c), Z_f)$, where $F$ is the set of video frames containing the object state. We select the top-ranked ones as they are most reliably recognized by the model. To further promote generalizability, we prioritize concepts that are shared across multiple object states. We enforce that at least half of the selected concepts are shared. Specifically, to select $k$ concepts, we first select the top-ranked shared concepts until $\lceil \frac{k}{2} \rceil$ are selected and then select remaining top-ranked concepts. In practice, we select the top 5 concepts for each object state and ensure that at least 3 of them are shared. We then construct a new graph with the selected visual concepts. An illustration of SAGE before and after refinement is shown in Figure 3.

## 3.3 Inference

We follow a two-step inference which first predicts a video-level action label, and then decode the object states with SAGE graph. The first step is skipped when the action is provided as input.

**Action Recognition:** We first identify the object of interest in the video. Each object $o$ is represented by the average embedding of all its associated states in SAGE. We then compute the similarity

between the object embedding and the embeddings of all video frames. The object of interest is selected as the one with the highest cumulative similarity to the video frames. Next, we predict the action in the video. For all actions potentially associated with the identified object based on the SAGE knowledge graph, we compute the similarity between each action embedding and the video-level embedding $Z_{\text{CLS}}$, and select the action $a^*$ with the highest similarity score.

**Object State Recognition:** For each frame $t$, we compute the similarities between its representation $Z_t$ and the state embeddings $R(s)$, and then a probability distribution over object states after softmax. The only difference is that we now narrow our scope of object states to only the initial, transitioning, and end states of the action $a^*$ instead of all the object states in SAGE. We then apply the Viterbi decoding over the state predictions to predict the object states following the temporal order in SAGE.

# 4 Experiment

## 4.1 Experimental Setup

**Datasets:** We evaluate our method on two object state recognition benchmarks, ChangeIt (Souček et al., 2022) and HowtoChange (Xue et al., 2024), which consist of videos depicting single actions. The task is to predict the physical state of the object of interest in every frame given the action. While SAGE can support state recognition from multi-action videos, by replacing the video-level action classification step with temporal action localization, to the best of our knowledge there is no publicly available benchmark with multiple objects and actions for evaluation purposes.

**Baselines:** We compare our method with LookForTheChange (LFC) (Souček et al., 2022), Multi-TaskChange (MTC) (Souček et al., 2024) and VidOSC (Xue et al., 2024), which are the state-of-the-art methods for object state recognition in single-action videos. LFC and VidOSC train separate specialized models for different objects / actions. MTC trains a unified model for all objects and actions given a known vocabulary. We also evaluate zero-shot VLMs, including CLIP (Radford et al., 2021), VideoCLIP (Xu et al., 2021), and InternVideo (Wang et al., 2022) for object state recognition.

To ensure a fair comparison, we use the same pre-trained VLM, CLIP ViT-L-14 (Radford et al., 2021), which is fine-tuned with the pseudo labels and videos from HowtoChange, as the vision backbone and pseudo-label generator for both the baselines and our method. The fine-tuning helps our method better recognize the objects and actions from videos, by utilizing noisy, automatically generated supervision from the speech modality, which we hope can improve the generalization performance at object state level. To understand the performance benefits when a better video-language model is used, we also train our model with VideoCLIP and compare it with the reported performance of the baselines (Xue et al., 2024; Souček et al., 2022, 2024) in Section 4.5.

**Evaluation Metrics:** Following Souček et al. (2022, 2024), we evaluate the methods in Precision@1 on the ChangeIt dataset. In this dataset, state precision refers to the precision of initial and end states, while action precision corresponds to the precision of transitioning states. We rename the metric as Trans. Pre@1 to avoid confusion. On the HowToChange dataset, we follow Xue et al. (2024) and evaluate the F-1 score, Precision, and Precision@1. See the Appendix for their detailed definition. While the object and action ground truth labels are available in the test set, they are optionally provided to the model according to the evaluation setup. The use of ground truth action and object annotation as privileged information is indicated in Table 1. When not provided, our method infers them directly from videos as described in Section 3.3. For the remaining tables, we follow the protocol from Xue et al. (2024) where action information is provided but the object label is not.

**Implementation:** We sample video frames at 1 FPS as in baselines (Xue et al., 2024; Souček et al., 2022, 2024). Frame embeddings are extracted by the vision encoder of the fine-tuned CLIP ViT-L-14 (Radford et al., 2021) (except in Section 4.5) and projected by a linear layer. We then process them using a three-layer Transformer (512-dimensional hidden states, four attention heads) followed by another linear projection layer to obtain frame representations. The action loss weight $\alpha$ is 0.1. We train the models using the AdamW optimizer with $1\mathrm{e}{-4}$ learning rate and $1\mathrm{e}{-4}$ weight decay. We use a batch size of 32 on ChangeIt and 128 on HowToChange. We train our model for 10 epochs on each dataset. These hyper-parameters are optimized according to validation performance. In the state-action graph construction, we use OpenAI `GPT-4o-mini-2024-07-18` as the LLM to generate 5 visual concepts for each object state. We also experiment with an open-weight model, Qwen3-32B, and find the performance differences are negligible compared to GPT-4o.

Table 1: Known object state recognition performance on ChangeIt and HowToChange. SAGE outperforms the baselines across different settings. The unified versions of baseline models suffer substantial performance degradation, whereas our model not only mitigates this issue but even surpasses specialized baseline models. *: Unified version implemented by us.

| Methods | Privileged Info Action | Object | Unified Model | ChangeIt State Pre@1 | Trans. Pre@1 | HowToChange F1 | Pre | Pre@1 |
|---|---|---|---|---|---|---|---|---|
| CLIP (Radford et al., 2021) | ✓ | ✓ | ✓ | 0.30 | 0.63 | 0.27 | 0.27 | 0.48 |
| VideoCLIP (Xu et al., 2021) | ✓ | ✓ | ✓ | 0.33 | 0.59 | 0.37 | 0.40 | 0.48 |
| InternVideo (Wang et al., 2022) | ✓ | ✓ | ✓ | 0.27 | 0.57 | 0.30 | 0.31 | 0.47 |
| LFC (Souček et al., 2022) | ✓ | ✓ | ✗ | 0.30 | 0.63 | 0.30 | 0.30 | 0.36 |
| SAGE (ours) | ✓ | ✓ | ✓ | **0.57** | **0.85** | **0.39** | **0.45** | **0.58** |
| VidOSC (Xue et al., 2024) | ✓ | ✗ | ✗ | 0.52 | 0.83 | 0.37 | 0.40 | 0.53 |
| SAGE (ours) | ✓ | ✗ | ✓ | **0.53** | 0.83 | 0.37 | **0.42** | **0.55** |
| LFC* | ✗ | ✗ | ✓ | 0.25 | 0.52 | 0.24 | 0.26 | 0.31 |
| VidOSC* | ✗ | ✗ | ✓ | 0.41 | 0.67 | 0.29 | 0.32 | 0.42 |
| MTC (Souček et al., 2024) | ✗ | ✗ | ✓ | 0.47 | 0.75 | 0.32 | 0.35 | 0.45 |
| SAGE (ours) | ✗ | ✗ | ✓ | **0.51** | **0.81** | **0.34** | **0.39** | **0.52** |

Table 2: Open-world evaluation results on ChangeIt and HowtoChange. The MTC method cannot generalize to novel actions or objects because its classification heads are fixed for the states of known objects. SAGE shows robust performance in the open-world setting, while baseline models degrade significantly on unseen actions or objects. *: Results from the best specialized models.

(a) Evaluation with known actions and novel objects.

| Methods | ChangeIt Novel Obj State Pre@1 | Trans. Pre@1 | HowtoChange Novel Obj F1 | Pre | Pre@1 |
|---|---|---|---|---|---|
| LFC* | 0.25 | 0.54 | 0.27 | 0.27 | 0.32 |
| VidOSC | 0.43 | 0.71 | 0.32 | 0.35 | 0.48 |
| SAGE (ours) | **0.49** | **0.78** | **0.34** | **0.39** | **0.50** |

(b) Evaluation with novel actions and novel objects.

| Methods | ChangeIt Novel Obj & Act State Pre@1 | Trans. Pre@1 | HowtoChange Novel Obj & Act F1 | Pre | Pre@1 |
|---|---|---|---|---|---|
| LFC* | 0.21 | 0.48 | 0.23 | 0.22 | 0.27 |
| VidOSC* | 0.27 | 0.59 | 0.25 | 0.28 | 0.37 |
| SAGE (ours) | **0.45** | **0.70** | **0.31** | **0.35** | **0.45** |

**Model Training and Time:** We train the model with $8\times$ NVIDIA V100 GPUs. It takes 30 minutes to extract visual embeddings for HowToChange and 3.5 hours for ChangeIt. The training takes about 30 minutes for HowToChange and 8 hours for ChangeIt.

## 4.2 Evaluation on Object State Recognition

We evaluate our method and baselines on both known and novel objects and actions. Prior work (Souček et al., 2022, 2024; Xue et al., 2024) trains separate specialized models for different objects or actions, and thus struggles to generalize to novel objects and actions. In this work, SAGE enables us to train a unified model for all objects and actions and generalize to novel objects and actions.

**Known Objects and Actions:** As shown in Table 1, naively extending baseline methods into unified models results in significant performance degradation. We reimplement the unified versions of the baselines by putting state prediction heads for all known actions and objects and on a shared backbone. With SAGE, our unified model with comparable number of parameters significantly outperforms the generalist baselines, and even outperforms the specialists. More importantly, while baseline models require the privileged information of actions and objects as inputs, our model can make precise object state recognition without knowing the action and object in the video during evaluation.

**Novel Objects and Actions:** We evaluate our model with novel objects as explored in prior work, and propose a more challenging setup where both the action and the object in the video are novel. Among the baseline methods, LFC and MTC cannot generalize to novel objects and actions because they rely on fixed-dimension classification heads trained specifically for seen object state transformations. Similarly, VidOSC cannot generalize to novel actions. To estimate their generalization ability and make a comparison, we follow Xue et al. (2024) and measure their performance upper bounds by enumerating all of their specialist models and pick the one with the best performance using the ground truth labels. Tables 2 reports the model performance on novel objects and novel actions. Our model maintains comparable performance on unseen objects and actions as it does on seen ones, whereas baseline models suffer significant performance drops.

Table 3: Parameter numbers and inference runtime of different methods on the HowtoChange dataset. When actions and objects are unknown, the specialized methods such as LFC and VidOSC must run all expert models and select the best one, which significantly increases the computational cost.

| Methods | #Params per Model | #Models | Runtime w/ Known Actions and Objects (s) | Runtime w/ Unknown Actions and Objects (s) |
|---|---|---|---|---|
| LFC (Souček et al., 2022) | 4.2M | 409 | 18.2 | 7278.4 |
| VidOSC (Xue et al., 2024) | 10.5M | 20 | 33.8 | 612.7 |
| MTC (Souček et al., 2024) | 8.1M | 1 | 24.5 | 24.5 |
| SAGE (Ours) | 10.9M | 1 | 29.7 | 29.7 |

Table 4: Comparison of SAGE before and after refinement on HowToChange.

| Graph | # Concepts | Known | | | Novel | | |
|---|---|---|---|---|---|---|---|
| | | F1 | Pre | Pre@1 | F1 | Pre | Pre@1 |
| SAGE (before refinement) | 5 | 0.37 | 0.42 | 0.55 | 0.31 | 0.35 | 0.45 |
| SAGE (before refinement) | 15 | 0.36 | 0.42 | 0.54 | 0.30 | 0.34 | 0.44 |
| SAGE (after refinement) | 5 | **0.38** | **0.42** | **0.58** | **0.33** | **0.39** | **0.50** |

**Efficiency Analysis:** We compare the parameter numbers and runtimes on the HowToChange dataset of our model and baselines. We compare their inference time on $8\times$ NVIDIA V100 GPUs. As shwon in Table 3, when the privileged information of actions and objects is not provided, our method show significant advantages in inference efficiency. Unlike specialized baselines that must run all specialized models and select the best one (following Xue et al. (2024)), our approach uses a single general model for all actions and objects, resulting in substantially reduced computational cost.

These findings demonstrate that our method enables the training of foundation models for object state recognition, paving the way for more scalable and generalizable solutions.

## 4.3 SAGE Graph Refinement

We evaluate our SAGE refinement strategy for both known and novel object states. To validate the effectiveness of our concept selection method, we compare the performance of original SAGE, SAGE with over-complete concepts and the refined SAGE with selected concepts. As shown in Table 4, the proposed refinement significantly improve the generalizability of SAGE.

In Figure 3, we illustrate the local SAGE graph structures of five object states before and after refinement. The refined graph incorporates more visual concepts that are shared across different object states, enabling the model to recognize novel object states by leveraging common concepts learned from known object states. Especially, before graph refinement, concepts that should be shared by multiple states might not be correctly assigned to these states (*e.g.*, "juice seep out" in Figure 3).

Table 5: Ablation studies on different technical designs in our method. We report the Pre@1 scores and their performance differences ($\triangle$) compared to the complete model.

| Methods | ChangeIt | | | HowToChange | | |
|---|---|---|---|---|---|---|
| | % Pre@1 ($\triangle$) | | | % Pre@1 ($\triangle$) | | |
| | Seen | Novel Obj | Novel Obj & Act | Seen | Novel Obj | Novel Obj & Act |
| W/o Textual Representation | 49.8 (-11.6) | N/A | N/A | 46.4 (-8.4) | N/A | N/A |
| W/o Visual Descriptions | 56.8 (-4.6) | 50.3 (-6.8) | 48.4 (-7.0) | 50.7 (-4.1) | 42.9 (-6.8) | 38.8 (-6.5) |
| W/o Joint Training | 62.3 (+0.9) | 54.6 (-2.5) | 47.9 (-4.1) | 55.5 (+0.7) | 48.2 (-1.5) | 41.9 (-3.4) |
| W/o Dynamic State Locating | 58.9 (-2.5) | 53.7 (-3.4) | 48.8 (-3.2) | 51.8 (-3.0) | 45.4 (-4.3) | 41.3 (-4.0) |

## 4.4 Ablation Studies

We analyze the effectiveness of our proposed method through ablation studies by removing each component of SAGE at a time. The results are shown in Table 5.

**State Text Embedding:** We remove the text embeddings of object states by not using the cosine similarity between language and vision representations for state recognition. Instead, we treat object states as discrete categories. The results indicate that removing textual representations significantly degrades model performance and disables it to work on unseen objects and actions.

Table 6: Evaluating SAGE with different VLMs. We observe that our approach is reasonably robust with respect to the choice of vision and text encoders.

| | ChangeIt | | HowtoChange | | |
|---|---|---|---|---|---|
| | State Pre@1 | Action Pre@1 | F1 | Pre | Pre@1 |
| SAGE + CLIP | 0.51 | 0.81 | 0.34 | 0.39 | 0.52 |
| SAGE + SigLIP | 0.53 | 0.82 | 0.34 | 0.42 | 0.56 |
| SAGE + MetaCLIP | 0.54 | 0.82 | 0.36 | 0.43 | 0.56 |

Table 7: Comparison between our model with VideoCLIP as pre-trained VLM and the reported results of baselines. Our model achieves slightly better performance than all baselines on seen objects and significantly outperform all baselines on novel objects.

| Methods | ChangeIt | | ChangeIt (Open-world) | | | | HowToChange | | | | | |
|---|---|---|---|---|---|---|---|---|---|---|---|---|
| | State Pre@1 | Action Pre@1 | State Pre@1 | | Trans. Pre@1 | | F1 (%) | | Precision (%) | | Pre@1 (%) | |
| | | | seen | novel | seen | novel | seen | novel | seen | novel | seen | novel |
| LFC Souček et al. (2022) | 0.35 | 0.68 | 0.36 | 0.25 | 0.77 | 0.68 | 30.3 | 28.7 | 32.5 | 30.0 | 37.2 | 36.1 |
| MTC (Souček et al., 2024) | 0.49 | 0.80 | 0.41 | 0.22 | 0.72 | 0.62 | 33.9 | 29.9 | 38.5 | 34.1 | 43.1 | 38.8 |
| VidOSC (Xue et al., 2024) | 0.57 | 0.84 | 0.56 | 0.48 | 0.89 | 0.82 | **46.4** | 43.1 | 46.6 | 43.7 | 60.7 | 58.2 |
| SAGE (Ours) | **0.60** | **0.89** | **0.59** | **0.55** | 0.89 | **0.87** | 46.4 | **44.7** | **47.5** | **46.3** | **63.6** | **61.2** |

**Visual Concept Descriptions:** We replace the visual concept descriptions with object state names to obtain the state embeddings, and keep the pseudo labels consistent for fair comparison. The results show that removing visual concepts leads to the largest performance drop on unseen objects and actions. This suggests that visual descriptions provide crucial reasoning cues for recognizing object states, especially when generalizing to novel objects and actions.

**Jointly Training with Action Recognition:** We remove the action recognition objective during training. Although this objective does not improve seen state recognition, it enhances the generalizability to unseen objects and actions. This is because unseen object states and actions may share similar relationships to seen ones, which could be learned by joint training with action recognition.

**Viterbi Temporal Decoding:** Instead of training the model with pseudo labels decoded with temporal constraints, we train it with pseudo labels generated by CLIP as in Xue et al. (2024). The results show that removing Viterbi decoding in pseudo label preparation leads to a significant drop in performance, suggesting that temporal constraints helps improve recognition, particularly for generalization.

**Vision-Language Models:** Finally, we explore the use of different vision-language models to showcase the robustness of our method. We adopt the visual and text encoders from SigLIP (Zhai et al., 2023), which enhances CLIP's training objective, and from MetaCLIP (Xu et al., 2024), which refines CLIP's training data quality. We evaluate in the most challenging setting where both the action and object information are unknown. In Table 6, we observe moderate improvements when SigLIP and MetaCLIP are used, indicating the image encoder is not the main performance bottleneck.

## 4.5   Comparison with Reported SOTA Results

For thorough comparison, we train our model by preplacing CLIP with VideoCLIP adopted by prior work, and compare SAGE with the reported results in baselines (Souček et al., 2022, 2024; Xue et al., 2024). Table 7 shows that our unified model can perform consistently better than the reported state-of-the-art results from specialized models.

## 4.6   Qualitative Results

We demonstrate in Figure 4 the top-1 frame predictions of our model for videos with known and novel objects and states. For each video, we identify the top-1 frame for the initial, transitional, and final object states by selecting the frame with the highest embedding similarity to the corresponding state. We also display the visual concepts with the highest embedding similarity to each top-1 frame. The results suggest that our model can accurately recognize the object states for both known and novel objects and actions.

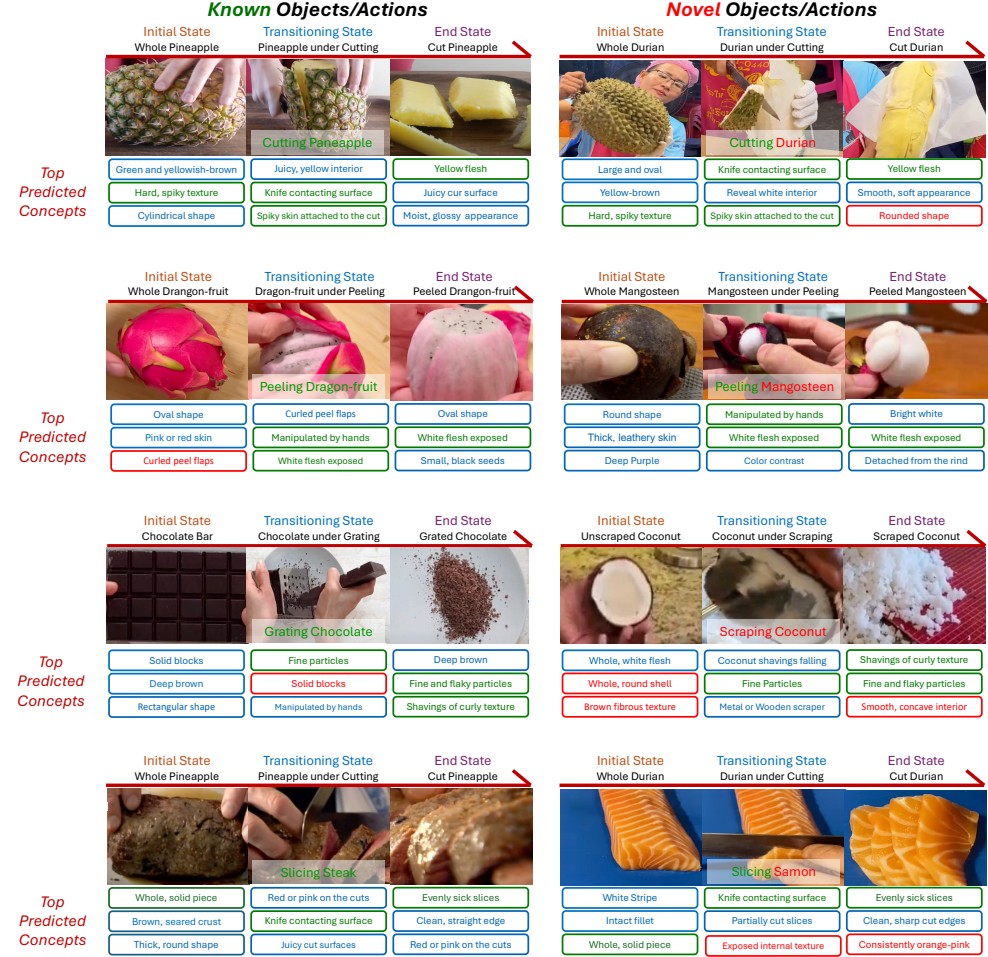

Figure 4: Examples of the top-1 frames and top-aligned visual concepts predicted by our model for the initial, transitional, and end states in the videos. In the action and object labels, text in *green* indicates known objects or actions and text in *red* indicates novel ones. In the visual concepts, the *green* concepts are shared across different object states while the *red* concepts are wrongly predicted.

## 5   Conclusion

In this paper, we propose a novel framework to build unified models for recognizing object states in videos. We leverage pre-trained LLMs and VLMs to build State-Action Graph Embeddings (SAGE) that decomposes object states into visual concepts. The graph structure where different objects and actions share the same visual concepts enables the model to generalize to novel objects and actions. Our model outperforms all baselines on two widely used object state transformation benchmarks, especially for open-world settings where both objects and actions are novel.

**Limitations:**   Our approach is evaluated on ChangeIt and HowToChange benchmarks, both contain a single action in each video. Although SAGE may be naturally generalized to handle videos with multiple non-overlapping actions via temporal action localization, it cannot directly support the scenario where multiple objects are undergoing state transformations *concurrently*.

We further envision two natural directions for future work: First, despite our effort to build a unified model for diverse objects and actions in different scenarios, our model is designed solely for object state and action recognition. Integrating the success of SAGE into a vision foundation model would be ideal as recognizing object physical state is a fundamental task of visual perception. Second, it would be highly impactful to apply our approach for downstream tasks such as tracking objects that undergo state transformations, or even temporal abstraction and skill discovery for robotic applications.

## Acknowledgments

We would like to thank Calvin Luo, Nate Gillman, Shijie Wang, Tian Yun, and Zilai Zeng for helpful discussions. This project was supported by Samsung. Our research was conducted using computational resources at the Center for Computation and Visualization at Brown University.

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

# A    Evaluation Metrics

On the ChangeIt dataset, we follow Souček et al. (2022, 2024) and report State Precision@1 for initial and end states, and Transition Precision@1 for the transitioning state. Note that the transition precision was referred to as *action* precision in prior works (Souček et al., 2022, 2024). We rename it here to avoid confusion. For each of the initial/transitioning/end states, the video frame with the top one probability predicted by a model is retrieved, where the precision is calculated as the percentage of corrected retrieved frames across all videos. Since the frame sampling strategy would affect the collection of candidate frames, we follow the same 1-FPS uniform sampling strategy as prior work so that the results are comparable.

Additionally, on the HowToChange dataset, we follow Xue et al. (2024) to evaluate the F-1 score, Precision, and Precision@1. The definition of Precision@1 is the same as Precision@1 in ChangeIt , except that the initial/transitioning/end states are jointly considered in this metric. Since Precision@1 only considers the top retrieved frames, F-1 score and Precision are used, both of which are computed over all sampled videos frames. We calculate the F-1 score and Precision for each of the initial, transitioning, and end states and report their average over the three states. Similarly, we use the same frame sampling strategy since the metrics are defined with respect to all sampled frames.

# B    SAGE Construction Details

We construct SAGE by querying the LLM to first provide the name of initial, transitioning and end states for an action and then provide the visual concepts for each state. In practice, we use OpenAI `GPT-4o-mini-2024-07-18`[1] as the LLM and add in-context examples in the prompts to guide it. In the following, we provide an instance of the prompts for generating the states and visual concepts. We skip the in-context examples below for conciseness.

**Prompt template for generating state names:**

```
Q: What are the initial, transitional, and end states of a(n) {object}
during the action {action}?
```

For example:

```
Q: What are the initial, transitional, and end states of a lemon during the
action slicing lemon?
A:
- whole lemon
- lemon under slicing
- sliced lemon
```

**Prompt for generating visual concepts:**

```
Q: What are the visual features for distinguishing a(n) {initial state
name}, a(n) {transitional state name} and a(n) {end state name}?
```

For example:

```
Q: What are the visual features for distinguishing a whole lemon, a lemon
under slicing and a sliced lemon?
A:
Whole lemon:
- spherical shape
- bright yellow color
- slightly dimpled texture
- smooth surface
- evenly distributed color
Lemon during slicing:
- manipulated by hands
```

---

[1] https://platform.openai.com/docs/models/gpt-4o-mini

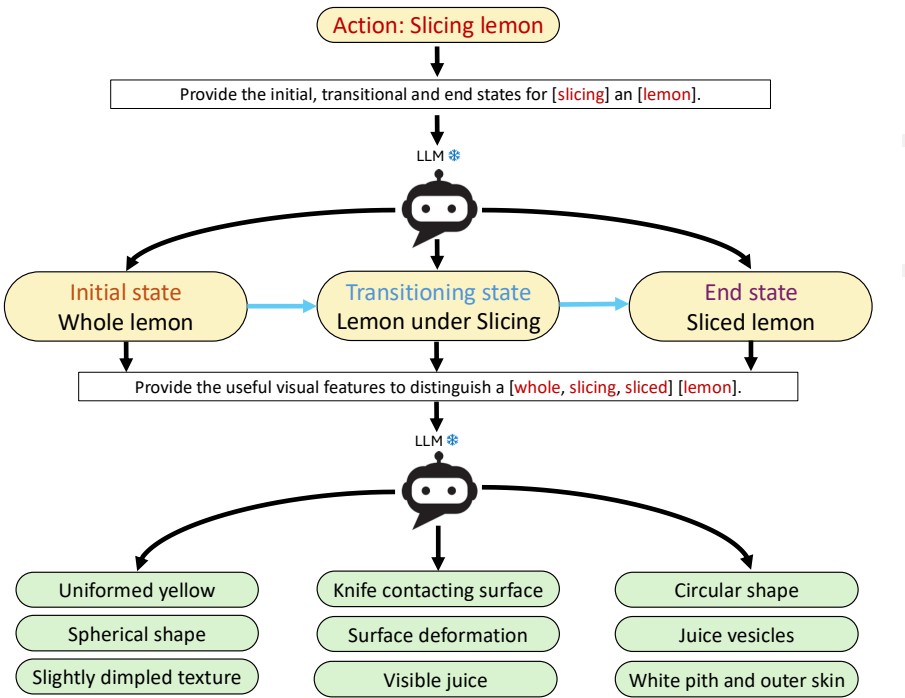

Figure A1: Illustration of how the state-action graph is constructed for a single action *slicing lemon* with a frozen LLM. See details in Section B.

```
- knife contacting surface
- surface deformation
- juice beginning to escape
- distinct cut line forming
Lemon after slicing:
- circular shape
- juicy interior
- white pith and outer skin
- exposed flesh glistening
- broken surface texture
```

We can obtain the object state names and visual concepts by parsing the LLM generated answers and collecting the visual concepts into a list. We observe that the answers consistently follow the in-context examples so individual concepts can be extracted by splitting over the dash ("-") sign. We further merge similar concepts according to the CLIP text embeddings. Two concepts are considered as similar if their cosine similarity over the text embeddings is higher than $0.9$. We merge greedily until no concepts have similarity higher than $0.9$. We use the pre-trained, frozen CLIP `ViT-L-14` text encoder, and normalize the extracted embeddings. As discussed in the method section, object state embeddings are calculated by averaging their corresponding visual concept embeddings. We normalize the state embedding again after taking the average.

## C  Dataset Details

We provide the details of the ChangeIt and HowToChange datasets in Table A8. These datasets consist of instructional videos for daily tasks beyond cooking (e.g., dyeing T-shirt). HowToChange contains a diverse set of objects, including a subset of novel objects that do not appear in the training set, making it suitable for evaluating the generalization ability of models. ChangeIt features a wider variety of actions and longer video sequences.

Table A8: Statistics of ChangeIt and HowToChange datasets.

| Datasets | #Objects | #Actions | #Videos | #Training | #Evaluation | Avg. Duration (s) |
|---|---|---|---|---|---|---|
| ChangeIt | 42 | 27 | 35,095 | 34,428 | 667 | 276 |
| HowToChange | 134 | 20 | 41,499 | 36,075 | 5,424 | 41 |

Table A9: Statistical significance of the object state recognition performance between our method and the baseline methods. We report p-values and statistical significance. Results marked with ✓ indicate statistical significance ($p < 0.05$).

| Methods | ChangeIt | | ChangeIt (Open-world) | | | | HowToChange | | | | | |
|---|---|---|---|---|---|---|---|---|---|---|---|---|
| | State Pre@1 | Action Pre@1 | State Pre@1 | | Trans. Pre@1 | | F1 (%) | | Precision (%) | | Pre@1 (%) | |
| | | | seen | novel | seen | novel | seen | novel | seen | novel | seen | novel |
| SAGE v.s. LFC Souček et al. (2022) | 6.1e-20 (✓) | 1.0e-20 (✓) | 4.1e-17 (✓) | 4.9e-29 (✓) | 5.4e-09 (✓) | 9.6e-17 (✓) | 1.3e-66 (✓) | 5.9e-67 (✓) | 3.1e-57 (✓) | 2.2e-68 (✓) | 1.9e-166 (✓) | 9.3e-151 (✓) |
| SAGE v.s. MTC (Souček et al., 2024) | 5.5e-05 (✓) | 5.6e-06 (✓) | 4.9e-11 (✓) | 3.2e-35 (✓) | 4.7e-15 (✓) | 1.1e-25 (✓) | 3.0e-40 (✓) | 3.5e-57 (✓) | 2.9e-21 (✓) | 2.1e-38 (✓) | 1.3e-101 (✓) | 2.2e-120 (✓) |
| SAGE v.s. VidOSC (Xue et al., 2024) | 2.7e-01 (✗) | 7.5e-03 (✓) | 2.7e-01 (✗) | 1.1e-02 (✓) | 1.0e+00 (✗) | 1.2e-02 (✓) | 1.0e+00 (✗) | 9.3e-02 (✗) | 3.5e-01 (✗) | 6.5e-03 (✓) | 1.8e-03 (✓) | 1.4e-03 (✓) |

# D  Statistical Significance

We conduct Students' T-test to evaluate the statistical significance of the result of our main experiments in Table 6. The results are shown in Table A9. Our model significantly outperforms LFC and MTC for both known and novel objects and significantly outperforms VidOSC for novel objects.

