# OpenReview forum: "SAGE: A Unified Framework for Generalizable Object State Recognition with State-Action Graph Embedding"
_NeurIPS.cc/2025/Conference — NeurIPS 2025 oral_

### Official Review · Reviewer_SfBi · 2025-06-25

**Clarity:** 2
**Significance:** 3
**Originality:** 3
**Rating:** 5
**Confidence:** 4

**Summary:**

The authors present SAGE (State-Action Graph Embedding), a framework for recognizing physical object state transitions in videos. The core idea is to create a unified model that can generalize to unseen objects and actions. This is achieved by decomposing object states into a set of fine-grained, language-described visual concepts. The framework leverages a Large Language Model (LLM) to construct an initial State-Action Graph, which is then refined using a Vision-Language Model (VLM). The resulting graph provides a compositional representation of states and actions, which is used to train a video transformer for the final recognition task.

**Questions:**

My main concerns are the following:
- Could the authors provide a more detailed explanation of the Vidertbi Decoding, ideally accompanied by a qualitative example?
- What is the methodology for validating the accuracy and, crucially, the visual relevance of the states and concepts generated by the LLM?
- The experiments rely on a single, powerful LLM. How dependent is the framework's overall performance on this specific choice?

For more details, please read the weaknesses section.

**Ethical Concerns:**

["NO or VERY MINOR ethics concerns only"]

**Final Justification:**

After reviewing the rebuttal and engaging in the discussion phase with the authors, I find that they have improved the clarity of the paper and satisfactorily addressed all of my concerns. Accordingly, I am raising my final recommendation to accept.

**Limitations:**

Yes

**Quality:**

2

**Strengths And Weaknesses:**

## Paper Strengths
- **Novel and Effective Conceptual Framework**: By representing states not as monolithic entities but as compositions of more primitive visual attributes, the proposed framework addresses the key challenge of generalization. This compositional approach forms a strong basis for the model's ability to recognize novel object states by leveraging familiar concepts learned from different contexts.
- **Comprehensive Empirical Evaluation**: The authors conduct extensive experiments and provide insightful ablation studies that convincingly validate the effectiveness of their approach.
    - *Performance*: As shown across different settings (Table 1), SAGE outperforms existing baselines on standard benchmarks.
    - *Proven Generalization*: The results in open-world settings (Table 2 (b)), where both objects and actions are unseen during training, are impressive and directly support the paper's main claims about generalizability.
    - *Efficiency*: The framework demonstrates a significant advantage in inference efficiency over specialist models (Table 3), which require running multiple experts when the action is unknown. This makes SAGE a more practical and scalable solution.
    - *Insightful Ablations*: The ablation studies clearly demonstrate the contribution of each component, particularly the value of using visual concept descriptions and the Viterbi temporal decoding.
- **Reproducibility and Clarity**: The authors' decision to release their code enhances the reproducibility of their work and provides a valuable resource.

## Paper Weaknesses
- **Insufficient Explanation of Viterbi Decoding**: The use of a constrained Viterbi decoding algorithm to refine the pseudo-labels is a key step in handling the lack of frame-level annotations. However, the explanation provided (L131-133) is very brief. While Figure 2 shows its place in the pipeline, its functional importance is understated. The paper would be significantly strengthened by a more detailed explanation accompanied by a qualitative example. For instance, a short walkthrough showing how the algorithm corrects an illogical, noisy label sequence (e.g., [Initial, End, Transitioning, End]) into a temporally coherent one would provide much-needed clarity and better highlight the value of this component.
- **Over-reliance on LLMs without Robustness Analysis**: The SAGE graph, the core of the framework, is constructed almost entirely by an LLM. This creates a critical dependency on the quality and consistency of the LLM’s output, yet this dependency is not examined or analyzed.
    - *LLM Sensitivity*: The authors rely on a single, powerful proprietary model (GPT-4) to generate states and concepts. Including an ablation study on the choice of LLM would be beneficial. How would the framework perform if a smaller, open-source model were used? Such an analysis would help assess the robustness of the approach and clarify how tightly its performance depends on a specific high-end LLM.
    - *LLM Output Validation*: The authors do not address how the correctness of the LLM-generated states and visual concepts is verified. Including a brief analysis of the quality and consistency of the generated graph would be interesting.

---

> ### Author Rebuttal · Authors · 2025-07-31
>
> Thank you for the thoughtful feedback. Below we provide detailed answers to your questions.
>
> ---
>
> ### Q1: Viterbi Decoding
> For each frame, the model predicts the probability of being in one of three states: initial, transitional, or end. To produce the most probable and temporally consistent sequence of states, we apply Viterbi decoding with a monotonic constraint: the sequence must follow the order initial → transitional → end. The pseudocode for the constrained Viterbi decoding is provided below. We will clarify Viterbi Decoding in the revised version and provide the full algorithm in Appendix.
>
> **Input**:
> - `T`: total number of frames
> - `S = {0: initial, 1: transitional, 2: end}`
> - `probs[t][s]`: probability of frame `t` being in state `s`
>
> **Initialization**:
> - `dp[0][0] ← log(probs[0][0])`
> - `dp[0][1] ← -∞`, `dp[0][2] ← -∞`
>
> **Recursion**:
> For `t = 1` to `T - 1`:
>  For `s in S`:
>   Let `prev(s)` be allowed previous states:
>    `prev(0) = {0}`
>    `prev(1) = {0, 1}`
>    `prev(2) = {1, 2}`
>   `dp[t][s] ← max_{p in prev(s)} [dp[t-1][p] + log(probs[t][s])]`
>   `back[t][s] ← argmax_{p in prev(s)} [dp[t-1][p] + log(probs[t][s])]`
>
> **Termination**:
> - `y_T ← argmax_s dp[T-1][s]`
>
> **Backtrace**:
> - For `t = T-2` to `0`:
>   `y_t ← back[t+1][y_{t+1}]`
>
> **Output**:
> - Sequence `y = [y_0, ..., y_{T-1}]` with enforced order: initial → transitional → end
>
> ---
>
> ### Q2: LLM Reliability
> To verify the reliability of LLM, we manually annotate the states and concepts generated by LLMs. We randomly sample 100 states and 100 concepts generated by the LLM and manually annotate the precision of these states and concepts. The precision of states is 100% and the precision of concepts is 92%, which shows that the LLM can generate highly accurate descriptions of object states and concepts.
>
> ### Q3: LLM Sensitivity
> To verify that our method is not sensitive to the choice of LLM, we conduct the experiments with another LLM, Qwen3-32B. We compare the performance of SAGE with GPT4 and Qwen3 under the settings of Table 1 and Table 2 on HowToChange dataset. SAGE demonstrates similar performance when combining with different LLMs, showing that our method is not sensitive to the choice of LLM.
>
> Table 1: Object state recognition
> | Methods       | Action | Object | Unified Model | F1   | Pre  | Pre@1 |
> |---------------|--------|--------|----------------|------|------|--------|
> | SAGE (GPT4)   | ✗      | ✗      | ✓           | 0.34 | 0.39 | 0.52 |
> | SAGE (Qwen3)   | ✗      | ✗      | ✓            | 0.34 | 0.40 | 0.52 |
>
> Table 2(a): Object state recognition for novel object
> | Methods       | F1   | Pre  | Pre@1 |
> |---------------|------|------|--------|
> | SAGE (GPT4)      | 0.34 | 0.39 | 0.50 |
> | SAGE (Qwen3)    | 0.35 | 0.39 | 0.49|
>
> Table 2(b): Object state recognition for novel object&action
>
> | Methods       | Action | Object | Unified Model | F1   | Pre  | Pre@1 |
> |---------------|--------|--------|----------------|------|------|--------|
> | SAGE (GPT4)  | 0.31 | 0.35 | 0.45 |
> | SAGE (Qwen3) | 0.29| 0.34| 0.44  |

---

> > ### Comment · Reviewer_SfBi · 2025-08-03
> >
> > I thank the authors for their response. My main concerns regarding the Viterbi decoding and the framework's dependence on a specific LLM have been fully addressed. The comparison of SAGE with GPT-4 and Qwen3 on the HowToChange dataset is particularly interesting. To make the comparison more complete, I encourage the authors to also include results on the ChangeIt dataset using Qwen3 in the final version.
> >
> > With my primary concerns resolved, I will raise my score to **Accept**.

---

> > > ### Author Response · Authors · 2025-08-08
> > > **Thanks to the reviewer's reply**
> > >
> > > We thank the reviewer for their insightful feedback and for raising the score. We appreciate the suggestion to include results on the ChangeIt dataset with Qwen3, and we will incorporate these results in the final version to provide a more complete comparison. The feedback has been invaluable in improving the clarity and completeness of our work.

---

### Official Review · Reviewer_LFZv · 2025-06-29

**Clarity:** 3
**Significance:** 2
**Originality:** 2
**Rating:** 3
**Confidence:** 4

**Summary:**

This paper introduced SAGE (State-Action Graph Embeddings), a framework that improved object state recognition in videos by breaking down states into detailed visual concepts. The proposed method combined large language models and vision-language models for better generalization to new objects and actions. SAGE outperforms existing methods by up to 14.6% in recognizing novel states while being more efficient.

**Questions:**

Please see the Weaknesses.

**Ethical Concerns:**

["NO or VERY MINOR ethics concerns only"]

**Limitations:**

Yes

**Quality:**

3

**Strengths And Weaknesses:**

Strengths:
1. SAGE outperforms existing methods by generalizing well to novel objects and actions, significantly improving object state recognition, even in open-world settings.
2. The framework combines vision-language models (VLMs) and large language models (LLMs), leveraging multimodal pre-trained knowledge for more accurate state recognition.
3. The framework allows for automatic addition of novel actions and objects, making it scalable to new data without extensive retraining.

Weaknesses:
1. Building the initial State-Action Graph (SAGE) requires the integration of multiple models, including pre-trained large language models (LLMs) and vision-language models (VLMs). This complex multi-step process of graph construction, followed by multimodal refinement, could make the system challenging to implement and maintain. Furthermore, the effectiveness of the framework heavily depends on the proper alignment between these models, which might not always be straightforward.
2. While SAGE can learn from unlabeled instructional videos, the performance of the system may still be dependent on the quality and diversity of the data used. In particular, if the video data lacks sufficient variation in object states or actions, the model may struggle to generalize to unseen scenarios, reducing its overall robustness. In some real-world applications, high-quality labeled data might not always be available, limiting the system's ability to perform effectively.
3. The framework’s reliance on multiple sophisticated models—such as LLMs and VLMs—requires substantial computational resources for both training and inference. The large-scale processing of multimodal data and the continuous refinement of the state-action graph could lead to increased memory usage and longer processing times, particularly in more complex scenarios or when dealing with large datasets. This could make it less suitable for enviroWhile SAGE is designed to generalize across novel objects and actions, there remains a risk of overfitting on certain specific objects or actions present in the training set. If the system is exposed to a narrow range of actions or objects during training, it might fail to generalize effectively when encountering new or highly diverse objects or actions in real-world settings. This overfitting issue could limit its scalability and applicability in dynamic, unpredictable environments.nments with limited computing power.
4. While SAGE is designed to generalize across novel objects and actions, there remains a risk of overfitting on certain specific objects or actions present in the training set. If the system is exposed to a narrow range of actions or objects during training, it might fail to generalize effectively when encountering new or highly diverse objects or actions in real-world settings. This overfitting issue could limit its scalability and applicability in dynamic, unpredictable environments.

---

> ### Author Rebuttal · Authors · 2025-07-31
>
> Thank you for the thoughtful feedback. Below we provide detailed answers to your questions.
>
> ---
>
> ### Q1: The integration of multiple models could make the system challenging to implement and maintain
>
> Integrating LLMs and VLMs is a common practice in multimodal learning, as it provides an **easy** approach to utilize the web-scale prior knowledge encoded by these models. Notably, all baseline approaches we compare against used VLMs to generate pseudo labels. In SAGE, the LLM is only responsible for proposing relevant visual concepts for each object state. The design is highly modular and not tied to other components in the overall framework, hence easy to implement. We invite the reviewer to check our actual implementation, which was included in the supplementary materials.
>
> Our system is also easy to maintain. Table 2 shows that our approach naturally generalizes to new actions and objects, without the need for re-training.
>
> ---
>
> ### Q2: Proper alignment between the models are needed.
>
> Alignment is not needed due to our modular design. As discussed in Section 3.2, the LLM is responsible for proposing an over-complete concept list, based on its “commonsense” knowledge. The concepts are then refined by a VLM based on how visually discriminative each concept is. As discussed in our response to reviewer LFZv, SAGE’s performance is robust with respect to the choice of the LLM. As shown in Table 1 and 6, SAGE achieves competitive performances when different VLMs are used, where as expected, better VLM leads to higher performance.
>
> ---
>
> ### Q3: "Performance dependent on quality and diversity of the data … high-quality labeled data might not always be available"
>
> As pointed out by reviewer LFZv themselves, “SAGE can learn from **unlabeled** instructional videos”, which allow us to utilize diverse and large-scale training data, without relying on labels.
>
> ---
>
> ### Q4: LLMs and VLMs require substantial computational resources for both training and inference
>
> The LLM is only used for graph construction at training time. The VLM is used for graph construction at training time, and visual feature extraction at training and test time. The use of the visual encoder in a VLM is a standard practice adopted by all baseline approaches.
>
> Once the visual representations are extracted, only a lightweight temporal Transformer is needed during inference time.
>
> ---
>
> ### Q5: "Overfitting on certain specific objects or actions present in the training set … If the system is exposed to a narrow range of actions or objects during training, it might fail to generalize effectively"
>
> As discussed earlier, our method does not rely on any labeled videos, hence can naturally utilize large-scale training data that covers a wide range of actions and objects. As shown in Table 2, our approach achieves the best performance on generalization towards novel actions and objects.

---

> > ### Comment · Reviewer_LFZv · 2025-08-07
> > **Thanks to the author's response**
> >
> > I appreciate the author's response, which addresses my concerns about the complexity of the model, the quality of the alignment of the modalities, and the diversity of the samples. I will therefore reconsider the final score.

---

> > > ### Author Response · Authors · 2025-08-08
> > > **Thanks to the reviewer's reply**
> > >
> > > We thank the reviewer for their constructive comments and for reconsidering their assessment after our clarifications. We will revise the paper to better highlight the modularity and ease of maintenance of our approach, clarify the role of each component, and emphasize how our design enables scalability and generalization to novel actions and objects.

---

### Official Review · Reviewer_hicx · 2025-07-02

**Clarity:** 1
**Significance:** 2
**Originality:** 2
**Rating:** 4
**Confidence:** 3

**Summary:**

This paper proposes a method for using LLMs and VLMs to recognize and classify object state changes in videos. The proposed SAGE (State-Action Graph Embeddings) first interprets each stage of the state change using short, natural language descriptions, then grounds these descriptions in vision via a pretrained VLM. The paper shows that this approach achieves good generalization to novel state changes, because these extracted visual concepts have compositional generalization capabilities.

**Questions:**

Please see the weakness section.

To summarize:
1. Clear explanation of what ground truth is available at test time, and exactly how we evaluate each metric.
2. Related to 1, is the current evaluation limited by groundtruth data format, and do we have any indication of state change temporal segmentation performance.
3. How does the performance compare with the groundtruth generation method?

I am open to increasing the score if these items are clarified.

**Ethical Concerns:**

["NO or VERY MINOR ethics concerns only"]

**Final Justification:**

Overall I do believe this paper is meaningful to the community with some imperfections, for example, the writing and presentation. My final rating is borderline accept.

**Limitations:**

Yes.

**Paper Formatting Concerns:**

None.

**Quality:**

2

**Strengths And Weaknesses:**

Strengths:
1. The work takes a generalist approach to state change modeling, which is more practically meaningful than a specialist approach.
2. The proposed approach has good interpretability, each state change recognition is supported by several visual concepts explained in natural language with associated probabilities.
3. The method achieves a considerable performance improvement, especially on novel actions.

Weaknesses:
1. Clarity. The paper is not particularly clear in a few places, especially settings, datasets, and evaluations. It seems that different datasets have different forms of annotations, and the paper should spend more time and space explaining them clearly.
2. The model in theory could distinguish fine-grained temporal segmentation of state changes (e.g., initial, transitioning, and end state). However, the experiments seem to only show state classification and action classification results. Is it because there is no ground truth for the former task of temporal segmentation?
3. Does the model obtain better performance on the final benchmarks than the pseudo ground truth generation method (using VLM)? Which experiments could serve as a good reference for that information?

---

> ### Author Rebuttal · Authors · 2025-07-31
>
> Thank you for the thoughtful feedback. Below we provide detailed answers to your questions.
>
> ---
> ### Q1: Clarity on evaluation setups
>
> At test time, we sample video frames at 1 FPS and feed them to the model. While the object and action ground truth labels are available in the test set, they are **optionally** provided to the model according to the evaluation setting. When the object and action labels are not provided, our model predicts them according to our method in Section 3.3. In Table 1, checkmarks in the Privileged Info column indicate whether object and action labels are provided. For Table 2, 4, 5, and 6, we follow the standard protocol from prior work (e.g. VidOSC), where action information is provided but the object label is not. Given the information above,the model predicts one of the following labels for each sampled video frame: initial, transitional, end, or background.
>
> To evaluate model performance, the test sets have frame-level annotations. Each label is one of  initial, transitional, end, or background. Following the implementation detail of evaluation metrics in Appendix A, we compute State Precision@1 and Transition Precision@1 on the ChangIt dataset and Precision, F-1 score, and Precision@1 on the HowToChange dataset. Note that we use different metrics on two datasets because previous works did so and we want to compare with them. This is **not** because two datasets have different forms of annotations.
>
> We will include these clarifications in the revised version.
>
> ---
> ### Q2: Why is temporal segmentation not evaluated?
>
> We follow the standard evaluation protocols as proposed by each of the original datasets. The original evaluation protocol was introduced by Alayrac et al. in ICCV 2017, where the authors described that “a temporal action localization is said to be correct if it falls within the ground truth time interval”. In practice, the metric is more robust in the presence of background frames that frequently occur in the videos.
>
> ---
>
> ### Q3: Pseudo ground truth generation method as a baseline?
>
> We would like to clarify that all approaches, including SAGE and the baselines, used pseudo ground truth generation as part of their pipelines. We note that there are two approaches where pseudo labels are used, as shown in Table 1:
> CLIP, VideoCLIP, and InternVideo are VLMs that directly predict pseudo labels on the **test videos**. This setup is usually referred to as “zero-shot” classification.
> SAGE, LFC, MTC, and VidOSC all use VLMs to generate pseudo labels on the **training videos**. These videos are then used to train machine learning models that make predictions on the test videos.
>
> To answer the reviewer’s particular question: If the goal is to understand how SAGE outperforms VLMs that generate pseudo labels on the test videos, then CLIP/VideoCLIP/InternVideo in Table 1 are the proper baselines.
>
> If the goal is to understand how SAGE outperforms models trained on videos with pseudo labels. Table 1/2/3 show the relevant comparisons, where we carefully control the pseudo label generator to be the same (CLIP ViT-L-14). Table 6 shows the comparison where the baselines can choose their own pseudo label generators (e.g. VideoCLIP, which was trained on HowTo100M).

---

> > ### Comment · Reviewer_hicx · 2025-08-04
> >
> > Thanks to the authors for the clarification. Please also include these in the next version of the paper. I keep my previous rating of borderline accept unchanged.

---

### Official Review · Reviewer_K9Kj · 2025-07-04

**Clarity:** 3
**Significance:** 3
**Originality:** 3
**Rating:** 4
**Confidence:** 2

**Summary:**

This paper proposes a unified object state recognition framework called SAGE.
The method is built upon state-action graphs generated by large language models (LLMs), which transform complex object states into a sequence of fine-grained, linguistically-describable visual concepts. These representations are then optimized multimodally using a vision-language model (VLM) to enhance generalization to novel objects and unseen actions. Finally, a video transformer decodes state changes from video frames, enabling efficient and broadly applicable state recognition.

**Questions:**

Please see the weaknesses.

**Ethical Concerns:**

["NO or VERY MINOR ethics concerns only"]

**Limitations:**

yes.

**Paper Formatting Concerns:**

Nothing.

**Quality:**

3

**Strengths And Weaknesses:**

Strengths:
- The paper is clearly written and easy to follow. The motivation is well-grounded, and the proposed method is simple and reproducible.
- The framework introduces a structured modeling approach through state-action graphs, effectively leveraging the synergy of language and vision to enable a highly interpretable and extensible state recognition system.

Weaknesses:
- Although the inference efficiency outperforms multi-model ensemble approaches, scalability may still be a challenge for extremely large action sets due to the need for exhaustive matching between concepts and states.
- The current evaluation is limited to single-action video scenarios, leaving the method's effectiveness in more complex multi-action settings (e.g., real-world videos) unclear.

---

> ### Author Rebuttal · Authors · 2025-07-31
>
> Thank you for the thoughtful feedback. Below we provide detailed answers to your questions.
>
> ---
>
> ### Q1: Scalability for extremely large action sets
>
> We would like to clarify that exhaustive matching between concepts and states is **not needed** during model inference. As illustrated in Figure 2, we first compute the embedding similarity to identify the video-level action label (unless the video-level action label is given as input) and then the object state recognition is restricted to the three object states and their corresponding concepts related to this action. We observe that recognizing video-level actions does not impose computational overhead since it only needs to be performed once per video (and the action space is moderate, since it is not combinatorial with states and concepts).
>
> ---
>
> ### Q2: Multi-action evaluation
>
> SAGE can technically be generalized to support multi-action videos by replacing the video-level action classification step with a temporal action localization / segmentation module.
> We agree that evaluations under multi-action settings would further demonstrate the effectiveness of our approach. Unfortunately, to the best of our knowledge there is no publicly available benchmark for recognizing object states from videos with multiple actions and objects. The most relevant existing dataset is Multiple Object States Transition (MOST) by Tateno et al., recently published in WACV 2025. This benchmark still assumes a single object is being manipulated within a video, but expands the list of object states to include those not strictly tied to actions (for example, an egg can be both “cracked” and “raw” after being cracked. In SAGE, we treat “raw” as a possible visual concept).

---

### Note · Authors · 2025-08-13

We thank all reviewers for their constructive feedback and for recognizing the novelty, scalability, generalizability, and interpretability of the SAGE framework. We also sincerely thank reviewers SfBi and LFZv for reconsidering their assessments and raising their scores after the rebuttal. We would like to highlight a few key responses to reviewer concerns and questions:

**Model complexity and efficiency (K9Kj, LFZv):** As clarified in our rebuttal, SAGE is modular, easy to implement, and incurs low training and inference costs. The LLM is only used once during training for concept generation, and inference relies on a lightweight temporal transformer.

**Data dependence and overfitting risk (LFZv):** SAGE can leverage large-scale unlabeled videos for self-supervised learning, removing reliance on annotated data. Our results show strong generalization to novel objects and actions.

**Clarity of methodology and evaluation (hicx, SfBi):** We provided detailed explanations of the evaluation setups, dataset protocols, and methodological steps. Specifically, we clarified the constrained Viterbi decoding algorithm with pseudocode, elaborated on the evaluation procedures, and explained the use of pseudo labels. These clarifications will be incorporated into the final version of the paper.

**LLM sensitivity and reliability (SfBi):** Experiments with GPT-4 and Qwen3 show consistent performance, confirming SAGE is not dependent on a specific LLM. Manual verification of LLM-generated concepts/states shows high precision (100% for states, 92% for concepts).

We will incorporate all clarifications, additional requested experiments, and detailed pseudo-label data in the final version to ensure transparency, reproducibility, and completeness. We believe these clarifications address the main concerns raised in the reviews and further highlight SAGE’s contributions as a practical, interpretable, and generalizable solution to object state recognition.

---

### Decision · Program_Chairs · 2025-09-17

**Decision:**

Accept (oral)

**Comment:**

This paper introduces a state-action graph embedding framework for recognizing the physical states of objects and their transformations within videos. All reviewers unanimously agree that it makes significant contributions and recommend acceptance. The AC notes that one reviewer (LFZv) stated the concerns had been resolved after the rebuttal but did not update the final rating. Accordingly, the AC has considered the reviewer’s comments rather than the unchanged rating.

After careful evaluation, the AC concurs with the reviewers’ assessments and agrees that this paper addresses a timely and promising direction by establishing the state-action graph for physical state recognition and transition, which is critical for many other applications such as neuro-symbolic reasoning and robotics. Its unified framework and strong empirical results make it broadly relevant and likely to attract a wide audience. Therefore, the AC recommends this paper be selected as an oral presentation.